# Korean Red Ginseng Ameliorates Allergic Asthma through Reduction of Lung Inflammation and Oxidation

**DOI:** 10.3390/antiox11081422

**Published:** 2022-07-22

**Authors:** Jin-Hwa Kim, Jeong-Won Kim, Chang-Yeop Kim, Ji-Soo Jeong, Je-Oh Lim, Je-Won Ko, Tae-Won Kim

**Affiliations:** 1Department of Veterinary Medicine (BK21 FOUR Program), Chungnam National University, 99 Daehak-ro, Daejeon 34131, Korea; jinhwa926@g.cnu.ac.kr (J.-H.K.); lilflflb@gmail.com (J.-W.K.); 963ckdduq@gmail.com (C.-Y.K.); jisooj9543@gmail.com (J.-S.J.); 2Department of Veterinary Medicine (BK21 FOUR Program), Chonnam National University, 77 Yongbong-ro, Buk-gu, Gwangju 500-757, Korea; dvmljo@gmail.com

**Keywords:** asthma, inflammation, Korean red ginseng, oxidative stress, ROS production

## Abstract

Six-year-old red ginseng, which is processed from the whole ginseng root via steaming and drying, has been shown to have preventive effects such as antioxidative, anti-inflammatory, and immunomodulatory. In this study, we evaluated the therapeutic effects of Korean red ginseng (KRG) against ovalbumin (OVA)-induced allergic asthma and the underlying mechanisms involved. We injected 20 µg of OVA on days 0 and 14, and mice were challenged with aerosolized OVA via a nebulizer for 1 h on days 21, 22, and 23. KRG was administered at 100 and 300 mg/kg from days 18 to 23. The KRG-treated mice showed significant reductions in their airway hyperresponsiveness, production of reactive oxygen species (ROS), and the number of inflammatory cells compared with the OVA-treated mice. The levels of type 2 cytokines in the bronchoalveolar lavage fluid and expression of OVA-specific immunoglobulin E in the serum, which were elevated in the OVA group, were reduced in the KRG-treated groups. The pro-inflammatory factors, inducible nitric oxide synthase and nuclear factor kappa-light-chain-enhancer of activated B cells, were downregulated by the KRG administration in a dose-dependent manner. KRG effectively suppressed the inflammatory response by inhibiting ROS production. Our results suggest that KRG may have the potential to alleviate asthma.

## 1. Introduction

Allergic asthma, one of the most common pulmonary diseases, is a complex response associated with various factors [1]. Studies have proven that the pro-inflammatory proteins, inducible nitric oxide synthase (iNOS), and nuclear factor kappa-light-chain-enhancer of activated B cells (NF-κB) play crucial roles in the pathogenesis of asthma [2], where they induce the production of T-helper (Th) 2 cytokines, especially interleukin (IL)-13, IL-5, and IL-4 [3]. The increased production of such Th2 cytokines sequentially increases inflammation, immunoglobulin (Ig) E production, and mucus overproduction, which are important phenotypes of asthma. Oxidative stress is another factor that is associated with the progression of asthma [4], with the increased levels of reactive oxygen species (ROS) inducing asthmatic responses via inflammatory activation signaling [5]. Excessive ROS production also promotes the antioxidative signaling molecules, including nuclear factor erythroid 2-related factor 2 (NRF2) and heme oxygenase-1 (HO-1) [6]. Therefore, focusing on the imbalance between the ROS expression and the levels of inflammatory and antioxidative molecules would be an important direction toward controlling asthma.

Ginseng Radix, which refers to the root part of ginseng, is one of the most widely used medicinal herbs in Korea [7]. When 6-year-old ginseng roots are harvested and processed through steaming and drying, they are named red ginseng [8]. One of the major pharmacological components of red ginseng are ginsenosides, which include Rb1, Rb2, and Rg3 [9]. According to previous studies, red ginseng has beneficial pharmacological effects, including antioxidative, anti-inflammatory, and immunomodulatory activities [7,10,11]. Additionally, red ginseng has been shown to reduce oxidative stress by decreasing ROS production in several cell lines, including gastric epithelial cells, the A549 cell line of hypotriploid alveolar basal epithelial cells, and the HEI-OC1 cell line of auditory cells [12,13,14]. In various studies, the antioxidative effect of red ginseng has also been demonstrated to result in the alleviation of the inflammatory response via the reduction of pro-inflammatory mediators, including NF-κB, mitogen-activated protein kinases, cyclooxygenase-2, and iNOS [15,16,17]. Furthermore, Lee et al. [18] reported that red ginseng can alleviate inflammation in the lungs and Guan et al. [19] found that ginsenoside Rg3 has ameliorative effects against chronic obstructive pulmonary disease exacerbation. In addition to the reports on the protective effect of red ginseng against these respiratory diseases, studies on its anti-asthmatic effect through its reduction of Th2 cytokines and IgE have also been published [20,21,22]. Based on these previous studies, further investigations of the protective effect of red ginseng against the onset of asthma and the underlying mechanisms of the oxidative stress-mediated inflammatory response are needed.

In this study, the anti-asthmatic effect of concentrated Korean red ginseng extract (KRG) was investigated by measuring airway hyperresponsiveness (AHR), inflammatory response, and mucus secretion in treated and untreated animals. Additionally, the underlying mechanisms associated with the KRG effects were evaluated by focusing our attention on the anti-inflammatory response mediated by the balance between ROS production and antioxidative activity.

## 2. Materials and Methods

### 2.1. High Performance Liquid Chromatography (HPLC)-Ultraviolet (UV) Analysis of Korean Red Ginseng

KRG (mixture of water extract from 75% of red ginseng taproots and 25% of rootlets) was obtained from the Korea Ginseng Corporation (Daejeon, Republic of Korea) and an analysis of its major ginsenoside marker components (viz., Rb1, Rd, and Rg3) was conducted utilizing an HPLC-UV detector (Agilent, Santa Clara, CA). Standard Rb1, Rd, and Rg3 compounds were purchased from Sigma-Aldrich (St. Louis, MO). The chromatographic separation was conducted using an Eclipse C_18_ column (3.5 μm, 2.1 × 150 mm; Agilent) at 40 °C. The mobile phases were distilled water (A) and acetonitrile (B), with the gradient conditions set at a flow rate of 1.2 mL/min as follows: 19–20% B, 0–10; 20–30% B, 10–28; 30–60% B, 28–40; 60–100% B, 40–48; 100–19% B, 48–49; and 19% B, 49–60 min. The injected sample volume was 5 μL and the UV detector was set at 203 nm.

### 2.2. Animals and Guidelines

Female BALB/c mice (six-week-olds) were acquired from Orient Co. (Seoul, Republic of Korea). All mice were housed under standard laboratory conditions (55 ± 5% humidity, 23 ± 2 °C, and 12 h of light/dark cycle) with ad libitum access to sterilized water and standard lab chow (Orient). Animal experiments were allowed by the Chungnam National University Animal Care and Use Committee (IACUC approval number: 202009A-CNU-142) and performed in accordance with established animal care, use, and experimental guidelines.

### 2.3. Experimental Procedure

The mice were randomly allocated to each group (5 groups; *n* = 6/group) after acclimatization, and OVA was then used to induce allergic asthma in 4 of the groups. In brief, each mouse was immunized with a 20 μg OVA (emulsified with 10 mg/mL of aluminum hydroxide and PBS) intraperitoneal injection on test days 0 and 14 for the sensitization of the allergen. On test days 21, 22, and 23, the mice were treated with aerosolized 1% (*w*/*v*) OVA in saline, which was administered 3 times a day for 20 min by an Omron nebulizer (Omron, Tokyo, Japan) for asthma progression. Dexamethasone (DEX) was used as the positive control drug. The 5 groups were designated as follows: NC (normal control; vehicle, p.o.), OVA (OVA treatment + vehicle, p.o.), DEX (OVA treatment + DEX 3 mg/kg, p.o.), KRG L (OVA treatment + KRG 100 mg/kg, p.o.), and KRG H (OVA treatment + KRG 300 mg/kg, p.o.).

### 2.4. Airway Hyperresponsiveness (AHR)

The degree of AHR of each mouse was assessed by whole-body plethysmography (Allmedicus, Seoul, Republic of Korea) at 24 h after the last OVA treatment. In brief, after 3 min of nebulization, which involved aerosolization with PBS followed by serial levels of methacholine (0, 10, 20, and 30 mg/mL), readings were obtained from the plethysmograph and averaged. The data are expressed in the enhanced pause (Penh) unit.

### 2.5. Analysis of Serum Immunoglobulin E and OVA-Specific Immunoglobulin E

At 24 h after completion of the AHR measurements, all animals were sacrificed (test day 25) and blood was gathered from the postcaval vein. The serum-containing supernatant was obtained after centrifugation at 800× *g* for 20 min. Then, the serum levels of IgE were determined utilizing an enzyme-linked immunosorbent assay (ELISA) kit (BioLegend, London, UK) and plate reader of ELISA (450 nm wavelengths: Bio-Rad Laboratories, Hercules, CA, USA) in accordance with the manufacturer’s guidelines.

### 2.6. Analysis of Bronchoalveolar Lavage Fluid

After blood collection, a tracheotomy was performed on each mouse and the left lung was infused with PBS (4 °C, 0.7 mL) with the right bronchus tied via an endotracheal tube and aspirated after a short while. The aspiration step was repeated to collect a total 1.4 mL volume of bronchoalveolar lavage fluid (BALF). After centrifugal separation of the BALF at 800× *g* for 5 min, the upper part was gathered and stored under −80 °C until Th2 cytokine analysis. The pellet of the remaining BALF cell was resuspended in 500 μL of PBS (4 °C) and the total cell number was calculated. Then, 5 × 10^3^ BALF cells were incubated with 20 µM of 2′,7′-dichlorofluorescin diacetate (Sigma-Aldrich, St. Louis, MO, USA) for 20 min at 37 °C. The quantitation of ROS activity was conducted using a plate reader of fluorescence (485 nm excitation and 530 nm emission wavelengths: PerkinElmer, Waltham, MA, USA). Then, 200 ul of the remaining resuspended BALF were attached to slides utilizing a cytospin device (Hanil Science Industrial, Seoul, Korea) for determining the differential cell counts. The slides were dried and the Diff-Quik stain was performed using Diff-Quik^®^ reagent (Sysmex Co., Kobe, Japan). The levels of Th2 cytokines were obtained with each detection kit (R&D System, Minneapolis, MN, USA) and a plate reader of ELISA (450 nm wavelengths: Bio-Rad Laboratories, Hercules, CA, USA).

### 2.7. Histopathological Analysis

After collection of the various samples of BALF, the left lung fixation was performed with 10% neutral-buffered formalin. After 1 week of fixation, each lung tissue was dehydrated, embedded, and sectioned using a HistoCore BIOCUT microtome (Leica Biosystems, Cambridge, UK). After the general dehydration process, the slides were stained with periodic acid-Schiff (IMEB Inc., San Marcos, CA, USA) to determine the degrees of mucus contents in the lungs. The slides were covered with mounting solution (Invitrogen, Carlsbad, CA, USA) and the data of quantitative analyses were obtained with an image analyzer (IMT i-Solution software, Huston, TX, USA).

### 2.8. Immunohistochemistry (IHC)

The iNOS (1:200 dilution; ab15323; Abcam, Cambridge, UK) and HO-1 (1:200 dilution; ab52947; Abcam) extent in the fixated tissue were determined utilizing IHC kits (Vector Laboratories, Burlingame, CA, USA) in accordance with the manufacturer’s indications. In brief, the tissues were sectioned, deparaffinized, dehydrated, and then antigen retrieval was performed with citrate buffer (pH 6.0) for 5 min at 121 °C. The slides were then sequentially applied with goat serum, primary antibodies, biotinylated secondary antibodies, and avidin–biotin–peroxidase complex. The protein expression levels were visualized using 3,3-diaminobenzidine and quantitated utilizing an image analyzer (IMT i-Solution software).

### 2.9. Western Blot Analysis

Using a BIOPREP-24R homogenizer (Allsheng Instruments Co., Hangzhou, China), the lung tissue was homogenized with cell lysis reagent (1/10, *w*/*v*; Sigma-Aldrich, St. Louis, MO, USA) included with an inhibitor cocktail of protease and phosphatase (Roche, Basel, Switzerland). To investigate the migration of NRF2 into the nuclei, cytosolic and nuclear fractions of the lung tissues were prepared using the previous method with minor modification [23], and the protein quantitation was calculated utilizing a BCA reagent (Thermo Fisher Scientific, Waltham, MA, USA). In accordance with a previous study, Western blot analysis was performed [24] using primary antibodies against the following proteins: NF-κB (1:1000 dilution; ab16502; Abcam), phosphor-NF-κB (1:1000 dilution; ab28856; Abcam), iNOS (1:1000 dilution; ab15323; Abcam), HO-1 (1:1000; ab52947; Abcam), NRF2 (1:500 dilution; ab92946; Abcam), α-tubulin (1:1000 dilution; sc8035; Santa Cruz Biotechnology, Sun Valley, Idaho), lamin-B1 (1:1000 dilution; sc374015; Biotechnology), and β-actin (1:2000 dilution; #4967; Cell Signaling Technology, Danvers, MA). After incubation, the membranes were washed three times, incubated with HRP-conjugated secondary antibodies (1:5000 dilution; LF-SA8001, LF-SA8002; Abfrontier Co., Ltd., Seoul, Republic of Korea) for 1 h at room temperature, washed three times, and then detected by enhanced chemiluminescence. For quantitation of the protein bands, their densitometric value was determined using a Chemi-Doc device (Bio-Rad Laboratories).

### 2.10. Statistical Analysis

All data are expressed as the mean plus standard deviation (SD). The statistical significance was confirmed using a one-way analysis of variance, followed by Dunnett’s multiple comparison. All statistical analyses were conducted using GraphPad InStat v3.0 software (La Jolla, CA, USA). A *p* value of less than 0.05 was considered to indicate statistical significance.

## 3. Results

### 3.1. HPLC-UV Analysis of Korean Red Ginseng

HPLC analysis of KRG was carried out with a UV detector for quantitative analysis of the major ginsenosides. The analytes were fully separated within 60 min, and a representative chromatogram is shown in Figure 1. The retention times for each ginsenoside were approximately 32.8 (Rb1), 34.5 (Rd), and 38.5 (Rg3) min. According to the optimized HPLC–UV analysis, the amounts of Rb1, Rd, and Rg3 in KRG were 6.7, 1.0, and 1.5 mg/g, respectively.

### 3.2. Effects of Korean Red Ginseng on Airway Hyperresponsiveness in Asthmatic Mice

The Penh value of the OVA group was significantly higher than that of the NC group at all methylcholine concentrations tested (10, 20, and 30 mg/mL) (Figure 2A). The KRG L and H groups administered the middle dose of methylcholine and the KRG H group administered the high dose of methylcholine all exhibited significantly lower Penh values than the OVA group. Although the KRG L group also showed a decrease in Penh values relative to the OVA group in response to the low and high doses of methylcholine, the differences in values were not statistically significant. The Penh value of the DEX group was significantly lower than that of the OVA group at all methylcholine concentrations tested.

### 3.3. Effects of Korean Red Ginseng on the Inflammatory Cell Counts in Bronchoalveolar Lavage Fluid from Asthmatic Mice

The OVA group showed significant increases in the number of inflammatory cells in BALF compared with the NC group (Figure 2B). In contrast, similar to the DEX group, the KRG L and H groups had significantly fewer inflammatory cells than the OVA group, particularly the eosinophils.

### 3.4. Effects of Korean Red Ginseng on Both Th2 Cytokine Production in Bronchoalveolar Lavage Fluid and IgE Levels in Serum from Asthmatic Mice

The BALF levels of IL-4 and IL-13 were significantly higher in the OVA group than in the NC group (Figure 3A–C; Table 1). In contrast, the KRG L and H groups displayed significantly lower IL-4, IL-5, and IL-13 levels than the OVA group. Although the KRG L group also had lower BALF levels of IL-5 than the OVA group, the difference was not statistically significant. These reductions in Th2 cytokine production levels were similar to those detected in the DEX group.

Similar to that of the Th2 cytokines, the production of total IgE and OVA-specific IgE was markedly elevated in the OVA group compared with that in the NC group (Figure 4; Table 1). However, compared with the OVA group, the KRG L and H groups had significantly lower total serum IgE and OVA-specific IgE levels. These reduction results were similar to those detected in the DEX group.

### 3.5. Effects of Korean Red Ginseng on Mucus Production in Lung Tissue from Asthmatic Mice

In lung tissue from mice, mucus overproduction (mucus positivity: cells stained dark purple) was significantly detected in the bronchial airways of mice in the OVA group compared with that of the animals in the NC group. However, such elevated levels of mucus production were significantly less evident in the KRG L and H groups (Figure 4). These reductions in mucus production were similar to those observed in the DEX group.

### 3.6. Effects of Korean Red Ginseng on Reactive Oxygen Species Production and Antioxidative Signaling Molecule Expression in Lung Tissue

The OVA group showed a significant increase in the production of ROS in the BALF compared with the NC group (Figure 5A and Appendix A). However, in the KRG L and H groups, there was a significant reduction in the ROS production level compared with that in the OVA group. Consistent with the ROS levels, the elevated expression of HO-1 detected in the OVA group was decreased in the KRG L and H groups in a dose-dependent manner, as revealed by Western blot and IHC analyses (IHC positivity: cells stained dark brown; Figure 5B–D). Similarly, NRF2 expression in the nuclei of the lung cells was increased in the OVA group, whereas it was markedly decreased in the KRG L and H groups, as revealed by the Western blot (Figure 6 and Appendix A).

### 3.7. Effects of Korean Red Ginseng on the Expression of iNOS and NF-κB in Lung Tissue

To further investigate the anti-inflammatory effects of KRG, the levels of iNOS and phosphorylated NF-κB p-65 expression were measured. As shown in Figure 7A,B, the expression levels of these inflammatory mediators were significantly increased in the OVA group, whereas these OVA-elevated levels were decreased in the KRG L and H groups in a dose-dependent manner (NF-κB: statistically significant in KRG H group, *p* < 0.01; iNOS: statistically significant in KRG L and H group, *p* < 0.01). Consistent with the Western blot results, IHC analysis revealed that the increased iNOS expression (IHC positivity: cells stained dark brown) level detected in lung tissue from mice of the OVA group was reduced in the animals of the KRG L and H groups (Figure 7C).

## 4. Discussion

In the present study, we evaluated the potential efficacy of KRG as an anti-asthmatic agent in an OVA-sensitized/challenged allergic asthma model, focusing on the oxidative stress and inflammatory responses in the animals. The results showed that KRG effectively suppressed the representative features of asthma, including the OVA sensitization/challenge-induced increases in the AHR, production of Th2 cytokines and IgE, inflammation, and mucus production. Additionally, KRG significantly suppressed OVA-induced ROS production, which consequently reduced the increase in HO-1 expression and migration of NRF2 into the nuclei. Moreover, KRG effectively inhibited the subsequent elevation of the iNOS and NF-κB expression induced by OVA sensitization/challenges in the mouse lung tissue.

Among the various factors related to asthma progression, the Th2 cytokines including IL-13, IL-5, and IL-4 play crucial roles in the advancement of the disease [25]. These cytokines not only accelerate the migration of eosinophils into the airways but also induce their maturation and activation, resulting in the elevation of other inflammatory cytokines and chemokines [26]. These latter molecules can cause airway obstruction by promoting the contraction of smooth muscle, inflammatory cell migration, and mucus secretion in the airways, which are the main features of asthma [27]. Therefore, the suppression of Th2 cytokines is a crucial strategy for the alleviation of asthma symptoms, with the efficacy of any test agent being predicated upon whether it can achieve this inhibitory effect. In the present study, KRG effectively reduced the number of OVA sensitization/challenge-induced inflammatory cells in BALF, especially the eosinophils. These results were accompanied by a decline in the production of IL-13, IL-5, and IL-4 in the BALF and total and OVA-specific IgE in the serum. This was in line with the previous research with the OVA asthma mice model [28]. These responses were consistent with the results of the AHR and histological findings, as revealed by the dose-dependent decreases in mucus secretion in lung tissues according to the amount of KRG administered. Taken together, our results strongly indicate that KRG significantly reduces asthma symptoms, such as inflammation and mucus hypersecretion, through the inhibition of Th2 cytokines.

An excessive increase in ROS levels is an important factor in the progression of asthma [29]. Increased ROS production in the airways not only directly induces the damage of various biological molecules but also leads to the release of histamine and mucus [30]. It was reported that ROS aggravates inflammation in the airway by increasing the permeability of inflammatory cells and relevant mediators, thereby inducing damage to the airway epithelial cells [31,32]. Generally, various biological molecules exist as antioxidants to respond to oxidative stress [33]. In particular, studies on the expression of direct antioxidant responsive element (ARE)-bearing antioxidants such as glutathione, superoxide dismutase, glutathione *S*-transferase, and catalase have been well reported in animal asthma mice models and clinical trials [34,35,36,37]. Among the indirect antioxidants, HO-1 is a sentinel signal in the diverse antioxidant-related gene regulated by NRF2 [38,39]. HO-1 is one such antioxidant that reduces oxidative stress by promoting the activation of other antioxidative molecules [40,41]. The expression of HO-1 is regulated by a series of transcription factors [42] such as NRF2. The implication of the NRF2/HO-1 axis against oxidative stress under asthma injury has been well demonstrated both in the laboratory as well as clinical researches [43,44,45]. Under oxidative stress conditions, NRF2 dissociates from Kelch-like epichlorohydrin-associated proteins and translocates to the nucleus [46], a process that increases HO-1 expression and triggers the antioxidative defense system [47]. Therefore, investigations of ROS production and antioxidative protein expression are important for elucidating the protective mechanisms of anti-asthmatic test agents. In this study, we conducted an experiment focusing on NRF2/HO-1 signaling rather than the well-known expression of direct ARE-bearing antioxidants. According to the results, KRG reduced the increase in ROS production induced by OVA sensitization/challenges in the lung tissue. Consequently, reduced expression of the antioxidants NRF2 and iNOS were observed in the KRG L and H groups, indicating that the mice administered the ginseng had developed the ability to inhibit ROS production. Ju et al. [48] reported that the protective effect of red ginseng against oxidative stress was due to its upregulation of antioxidative proteins. Another study reported that red ginseng reduced oxidative stress through the inhibition of ROS production [49]. Our results suggest that KRG mediates its protective effect against asthma progression by reducing ROS production rather than upregulating the antioxidants. Due to the different features of the diverse secondary metabolites of Korean red ginseng, including their biological potency, pharmacokinetic values, bioavailability, and elimination rates, it might be too speculative to indicate the leading active components of the presently explored KRG. However, although the animal models tested were different, the ginsenosides Rb1 and Rg3 were reported to activate the NRF2/HO-1 signaling pathways against pathologic stimuli [50,51]. Further time-course studies with pharmacokinetic analysis can offer supportive information on the recognized active components and underlying protective mechanisms in KRG against the allergic asthma model.

Inflammatory responses through the activation of various inflammatory signaling pathways are a well-known feature of asthma progression [52], with NF-κB being one of the transcription factors that induces this process [53]. The role of iNOS, an enzyme regulated by NF-κB, has been well established in many previous studies [54]. NF-κB activation upregulates the expression of iNOS, resulting in the elevation of nitric oxide production, which results in oxidative stress and inflammatory responses [55]. The relationship of NF-κB with the pathogenesis of asthma has been well proven in various animal studies and clinical trials including OVA-induced asthma mice models [28,56,57,58]. Furthermore, it has been shown that iNOS expression increases concurrently with the increased NF-κB expression in patients with asthma [59]. Given that several studies have evaluated the expression of NF-κB and iNOS to investigate the potential protective effects that natural products may have, it is clear that the modulation of these signaling molecules is an important strategy for asthma treatment.

In this study, the levels of NF-κB phosphorylation and iNOS expression were concurrently significantly increased in the OVA group, but the administration of KRG significantly reduced these OVA-induced effects in the mice in a dose-dependent manner. Our results indicate that one of the mechanisms involved in the KRG alleviation of inflammatory responses in asthma is to decrease the phosphorylation of NF-κB and the subsequent expression of iNOS.

## 5. Conclusions

In conclusion, KRG was found to alleviate the representative features of asthma including AHR, airway inflammation, and mucus secretion. These protective effects of KRG were closely related to its prevention of the depletion of antioxidative proteins, including NRF2 and HO-1, through the inhibition of ROS production. Concurrently, KRG reduced the inflammatory response by suppressing NF-κB phosphorylation and subsequently iNOS expression. Taken together, our results suggest that KRG can be a potent therapeutic agent for the treatment of asthma.

## Figures and Tables

**Figure 1 antioxidants-11-01422-f001:**
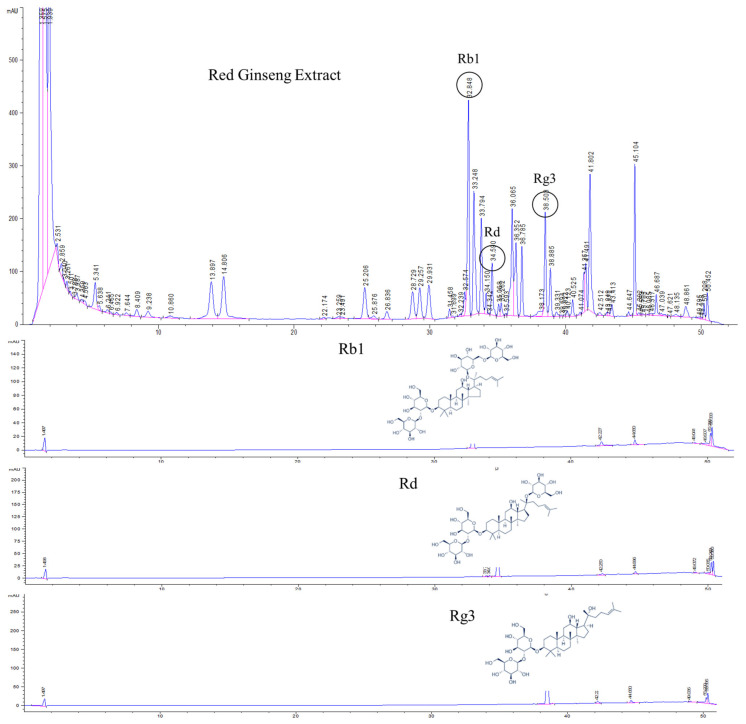
High-performance liquid chromatography (HPLC) analysis of Korean red ginseng (KRG) was carried out with an ultraviolet (UV) detector for quantitative analysis of the major ginsenosides. The 20(*S*)-protopanaxadiol ginsenoside Rb1, 20(*S*)-protopanaxadiol ginsenoside Rd, and 20(*S*)-protopanaxatriol ginsenoside Rg3 were detected.

**Figure 2 antioxidants-11-01422-f002:**
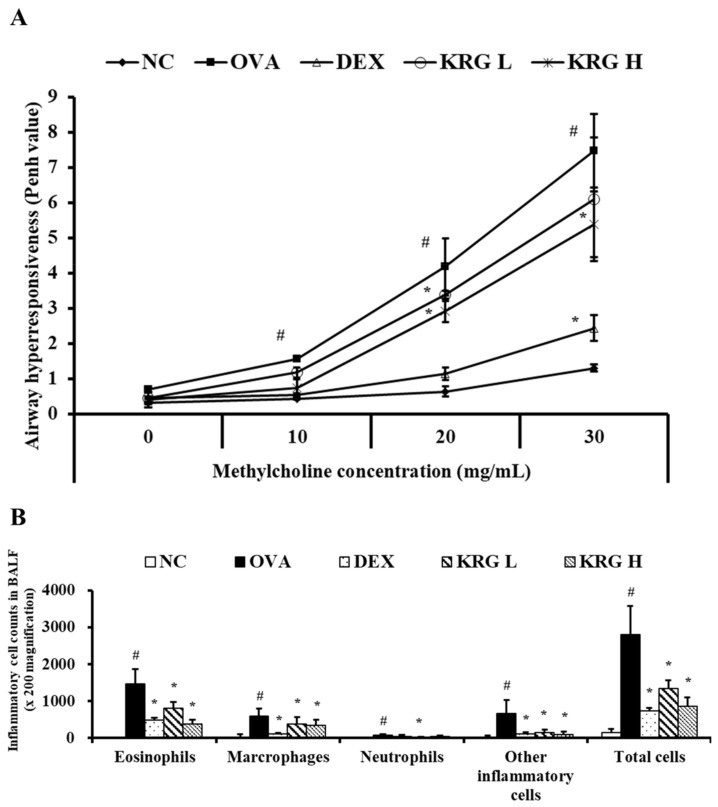
Korean red ginseng (KRG) decreases the elevated (**A**) airway hyperresponsiveness and the (**B**) inflammatory cell count in bronchoalveolar lavage fluid (BALF). Normal control (NC) group; mice administered with the vehicle, ovalbumin (OVA), group; mice treated with OVA and administered with the vehicle, dexamethasone (DEX), group; mice treated with OVA and administered with dexamethasone (3 mg/kg); KRG low (KRG L) and high (KRG H) groups; mice treated with OVA and administered with KRG (100 and 300 mg/kg, respectively). Values: means ± standard deviation (*n* = 6). Significance: ^#^
*p* < 0.05 vs. NC; * *p* < 0.05 vs. OVA.

**Figure 3 antioxidants-11-01422-f003:**
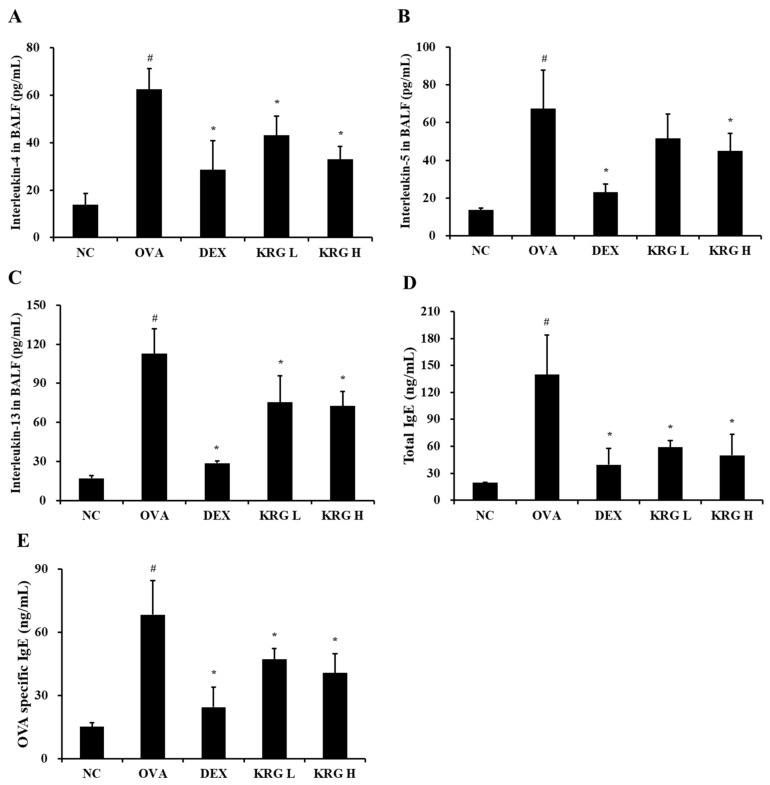
Korean red ginseng (KRG) reduces the elevated levels of interleukin (IL) -4 (**A**), IL-5 (**B**), and IL-13 (**C**) in the bronchoalveolar lavage fluid (BALF) and total immunoglobulin E (IgE) (**D**) and ovalbumin (OVA)-specific IgE (**E**) in the serum. Normal control (NC) group; mice administered with the vehicle, OVA, group; mice treated with OVA and administered with the vehicle, dexamethasone (DEX), group; mice treated with OVA and administered with dexamethasone (3 mg/kg); KRG low (KRG L) and high (KRG H) groups; mice treated with OVA and administered with KRG (100 and 300 mg/kg, respectively). Values: means ± standard deviation (*n* = 6). Significance: ^#^
*p* < 0.05 vs. NC; * *p* < 0.05 vs. OVA.

**Figure 4 antioxidants-11-01422-f004:**
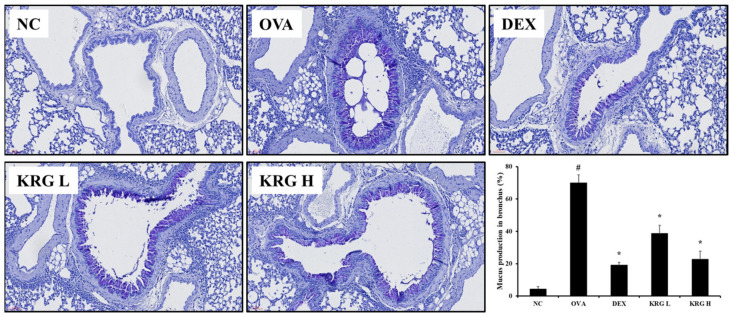
Korea red ginseng (KRG) inhibits the mucus overproduction in the lung tissue induced by ovalbumin (OVA) treatment. Quantitative analyses of mucus production were performed utilizing an image analyzer. Normal control (NC) group; mice administered with the vehicle, OVA, group; mice treated with OVA and administered with the vehicle, dexamethasone (DEX), group; mice treated with OVA and administered with dexamethasone (3 mg/kg); KRG low (KRG L) and high (KRG H) groups; mice treated with OVA and administered with KRG (100 and 300 mg/kg, respectively). Values: means ± standard deviation (*n* = 6). Significance: ^#^
*p* < 0.05 vs. NC; * *p* < 0.05 vs. OVA.

**Figure 5 antioxidants-11-01422-f005:**
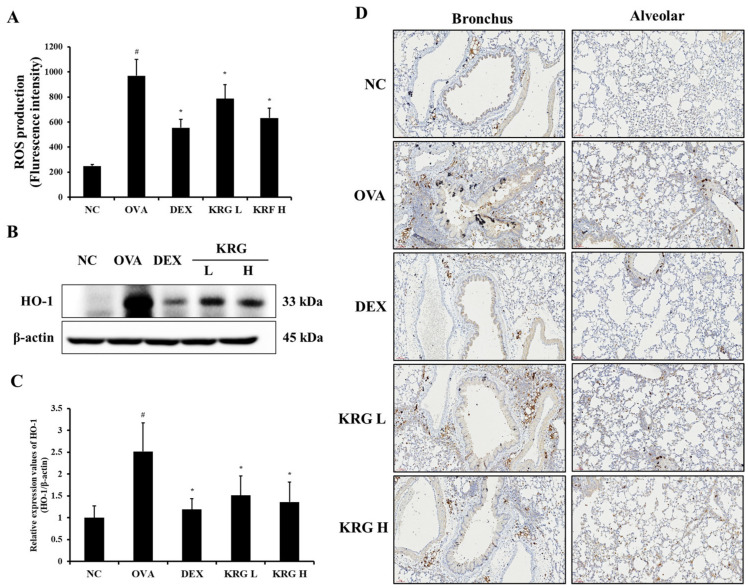
Korean red ginseng (KRG) reduces the levels of reactive oxygen species (ROS) production (**A**) in the bronchoalveolar lavage fluid (BALF) and expression of heme oxygenase-1 (HO-1) (**B**) in the lung tissues, of which (**C**) densitometric values were determined utilizing Chemi-Doc. (**D**) HO-1 expression in the bronchus and alveolar regions of lung tissues was revealed by immunohistochemistry (IHC). Normal control (NC) group; mice administered with the vehicle, ovalbumin (OVA), group; mice treated with OVA and administered with the vehicle, dexamethasone (DEX), group; mice treated with OVA and administered with dexamethasone (3 mg/kg); KRG low (KRG L) and high (KRG H) groups; mice treated with OVA and administered with KRG (100 and 300 mg/kg, respectively). Values: means ± standard deviation (*n* = 6). Significance: ^#^
*p* < 0.05 vs. NC; * *p* < 0.05 vs. OVA.

**Figure 6 antioxidants-11-01422-f006:**
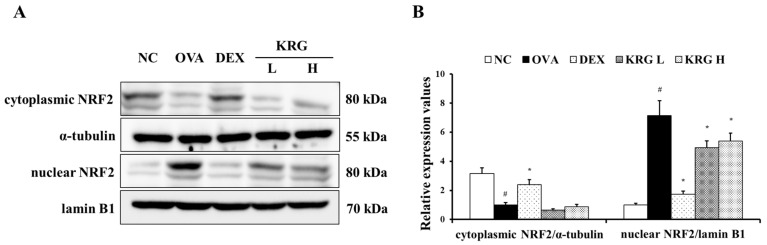
Korean red ginseng (KRG) reduces the translocation of nuclear factor erythroid 2-related factor 2 (NRF2) into the nuclei (**A**) in the lung tissue induced by ovalbumin (OVA) treatment. Densitometric values (**B**) were determined utilizing Chemi-Doc. Normal control (NC) group; mice administered with the vehicle, OVA, group; mice treated with OVA and administered with the vehicle, dexamethasone (DEX), group; mice treated with OVA and administered with dexamethasone (3 mg/kg); KRG low (KRG L) and high (KRG H) groups; mice treated with OVA and administered with KRG (100 and 300 mg/kg, respectively). Values: means ± standard deviation (*n* = 6). Significance: ^#^
*p* < 0.05 vs. NC; * *p* < 0.05 vs. OVA.

**Figure 7 antioxidants-11-01422-f007:**
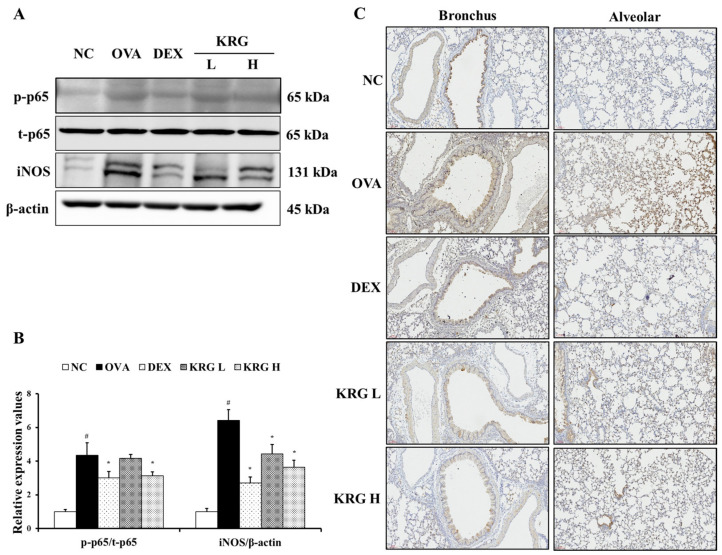
Korean red ginseng (KRG) reduces the levels of phosphorylated nuclear factor kappa-light-chain-enhancer of activated B cells (NF-κB) and inducible nitric oxide synthase (iNOS) expression in the lung tissue induced by ovalbumin (OVA) treatment. Images of band (**A**) were selected according to each protein’s molecular size and the relative ratios of each protein (**B**) were determined using Chemi-Doc. (**C**) iNOS expression in the bronchus and alveolar regions of lung tissues was revealed by immunohistochemistry (IHC). Normal control (NC) group; mice administered with the vehicle, OVA, group; mice treated with OVA and administered with the vehicle, dexamethasone (DEX), group; mice treated with OVA and administered with dexamethasone (3 mg/kg); KRG low (KRG L) and high (KRG H) groups; mice treated with OVA and administered with KRG (100 and 300 mg/kg, respectively). Values: means ± standard deviation (*n* = 6). Significance: ^#^
*p* < 0.05 vs. NC; * *p* < 0.05 vs. OVA.

**Table 1 antioxidants-11-01422-t001:** Levels of interleukins (IL) and immunoglobulin E (IgE) in bronchoalveolar lavage fluid or serum.

Product (pg/mL)	NC	OVA	DEX	KRG L	KRG H
IL-4	14.0 ± 4.71 ^a^	62.6 ± 8.65 ^#^	28.7 ± 12.08 *	43.1 ± 8.20 *	33.0 ± 5.41 *
IL-5	13.8 ± 1.01	67.5 ± 20.40 ^#^	23.0 ± 4.28 *	51.5 ± 12.92	44.9 ± 9.41 *
IL-13	17.2 ± 1.99	112.7 ± 19.06 ^#^	28.7 ± 1.86 *	75.4 ± 20.31 *	72.7 ± 11.13 *
Total IgE	19.3 ± 0.42	139.8 ± 44.11 ^#^	39.4 ± 18.37 *	59.0 ± 7.20 *	50.0 ± 23.61 *
OVA specific IgE	15.2 ± 1.97	68.4 ± 16.34 ^#^	24.5 ± 9.59 *	47.2 ± 5.23 *	40.7 ± 9.26 *

Normal control (NC) group; mice administered with the vehicle, ovalbumin (OVA), group; mice treated with OVA and administered with the vehicle, dexamethasone (DEX), group; mice treated with OVA and administered with dexamethasone (3 mg/kg); KRG low (KRG L) and high (KRG H) groups; mice treated with OVA and administered with KRG (100 and 300 mg/kg, respectively). ^a^ Values are presented as mean ± standard deviation (*n* = 6). ^#^ Significantly different from NC (*p* < 0.05). * Significantly different from OVA (*p* < 0.05).

## Data Availability

Data is contained within the article.

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
