# Peer review of "Korean Red Ginseng Ameliorates Allergic Asthma through Reduction of Lung Inflammation and Oxidation"

_antioxidants, 2022, doi:10.3390/antiox11081422_

Round 1
Reviewer 1 Report
The manuscript "Korean Red Ginseng Ameliorates Allergic Asthma through 2 Sparing the Depletion of Nrf2/HO-1 Signaling" is of great interest to the scientific community.
The experimental design is well structured and the methods have a clear but incomplete description. The results support the conclusions, however there are some speculations that need clarification. In the discussion it is essential to introduce current manuscripts with very similar work and discuss the results obtained.
In the material and methods section it is essential to correctly identify how Korean Red Ginseng was obtained and how it was used. It is an aqueous extract, an ethanolic extract,? It was in powder form and was solubilized in what proportion and with what?
The constitution of KRG was characterized by HPLC, but what proportion was injected? What correlation is there between this analysis and the 2 quantities used in the study? Why were 100 and 300 mg of KRG chosen. There are other papers, with the same type of experimental model and with smaller amounts.
Authors should reference "Korean Red Ginseng affects ovalbumin-induced asthma by modulating IL-12, IL-4, and IL-6 levels and the NF-kB/COX-2 and PGE2 pathways" Soon-Young Lee at al" J Ginseng Res 45 (2021) 482e489 and compare their results in the discussion.
In the section of 2.10. "Statistical Analysis", the authors repeat the text from the previous section.
In general the graphs have low graphical quality which makes interpretation difficult, e.g. fig 1C.
Author Response
Reviewer #1: The experimental design is well structured and the methods have a clear but incomplete description. The results support the conclusions, however there are some speculations that need clarification. In the discussion it is essential to introduce current manuscripts with very similar work and discuss the results obtained.
In the material and methods section it is essential to correctly identify how Korean Red Ginseng was obtained and how it was used. It is an aqueous extract, an ethanolic extract,? It was in powder form and was solubilized in what proportion and with what?
- We totally agree with reviewer’s indication. The concentrated Korean red ginseng extract (KRG) was obtained from KGC (Korea Ginseng Corporation). The M&M section was amended accordingly with the compound information.
The constitution of KRG was characterized by HPLC, but what proportion was injected? What correlation is there between this analysis and the 2 quantities used in the study? Why were 100 and 300 mg of KRG chosen. There are other papers, with the same type of experimental model and with smaller amounts.
- Thanks for your kind comments. The ginsenoside contents in the KRG was determined by HPLC analysis and the 5 uL of KRG was injected into the HPLC for column separation. Based on the sum of major ginsenoside contents we found which was less than total 10 mg/g, we set the dose range for the animal experiment to 100 and 300 mg/kg. We confirmed the safety of the dose hired with preliminary study.
Authors should reference "Korean Red Ginseng affects ovalbumin-induced asthma by modulating IL-12, IL-4, and IL-6 levels and the NF-kB/COX-2 and PGE2 pathways" Soon-Young Lee at al" J Ginseng Res 45 (2021) 482e489 and compare their results in the discussion.
- We largely agree with reviewer’s indication. We cited indicated references with the comments in the discussion part (2 and 4 paraphraphs).
In the section of 2.10. "Statistical Analysis", the authors repeat the text from the previous section.
- Thanks for your kind comments. We corrected the fatal mistakes that occurred in the process of changing the manuscript format according to Journal policy.
In general the graphs have low graphical quality which makes interpretation difficult, e.g. fig 1C.
- Thanks for your kind comments. We modified all the figures for better interpretation, and brief descriptions of the positive sign of the PAS and IHC stain have been inserted in the manuscript (section of 3.5 and 3.6)
Reviewer 2 Report
The manuscript “Korean Red ginseng ameliorates allergic asthma through sparing the depletion of Nrf2/HO-1 signaling” show that in the model of mice, Korean Red Ginseng downregulates inflammatory parameters in asthmatic mice possibly through Nrf2/HO1 signaling.
The manuscript has several flaws that should be corrected before making any decision.
Define the catalogue numbers for primary antibodies, and define secondary antibodies, and their dilution (Manufacturer and Cat.No.).
Under Chapter 2.10. Statistical Analysis the authors duplicated the Western blot method description.
The graphs are too small, and their quality is low. It is hard to see what the authors wanted to present.
The photos of IHC are rather small and it would be better to show higher magnifications. Indicate in the photos what is shown and the differences.
ON Western blots, molecular weights should be indicated. Also, blots with protein markers should be in supplemental data.
How do authors explain the unspecific bands for all blots? NRF2 and HO-1 are only one band without so many unspecific bands that the authors presented.
Author Response
Reviewer #2: The manuscript “Korean Red ginseng ameliorates allergic asthma through sparing the depletion of Nrf2/HO-1 signaling” show that in the model of mice, Korean Red Ginseng downregulates inflammatory parameters in asthmatic mice possibly through Nrf2/HO1 signaling.
The manuscript has several flaws that should be corrected before making any decision.
Define the catalogue numbers for primary antibodies, and define secondary antibodies, and their dilution (Manufacturer and Cat.No.).
- Thanks for your kind comments. We added additional information you pointed out in materials and methods section.
Under Chapter 2.10. Statistical Analysis the authors duplicated the Western blot method description.
- Thanks for your kind comments. We corrected the fatal mistakes that occurred in the process of changing the manuscript format according to Journal policy.
The graphs are too small, and their quality is low. It is hard to see what the authors wanted to present.
- Thanks for your kind comments. We have made general modifications to most of the graphs and photos of the figures to make them more readable.
The photos of IHC are rather small and it would be better to show higher magnifications. Indicate in the photos what is shown and the differences.
- Thanks for your kind comments. We increased the size of photos of IHC. Also, the resolution of photos has been increased so that it can be seen clearly even when magnified. In addition, definition of positive findings of IHC and PAS stain were added to the results section for better interpretation.
ON Western blots, molecular weights should be indicated. Also, blots with protein markers should be in supplemental data.
- Thanks for your kind comments. We inserted the molecular weights of proteins investigated in the present study. In addition, the original Western blot band submitted in the first submission process were inserted as supplemental data.
How do authors explain the unspecific bands for all blots? NRF2 and HO-1 are only one band without so many unspecific bands that the authors presented.
- Thanks for your kind comments. It is thought that there are various reasons for the occurrence of unspecific bands for western blot. According to our experimental experience so far, the 5 major reasons for unspecific band are as follows: inadequacy of tissue sampling process, buffer composition, pH, antibody purity, and blocking. These reasons may cause not only the generation of an unspecific band but also the splitting of the one target band into several strands at the expected molecular weight. Also, while there are antibodies that show these phenomena well, there are antibodies that show only the target band well (actin or HO-1 you pointed out). In the present study, unspecific bands appeared because all these factors could not be controlled.
However, it is not considered inappropriate to interpret a band that appears clearly at the molecular weight point suggested by the manufacturer as a target protein. We hope that our answer will be an appropriate response to your comments.
Reviewer 3 Report
Revision - manuscript titled: “Korean Red Ginseng Ameliorates Allergic Asthma through Sparing the Depletion of Nrf2/HO-1 Signaling”
The authors have extensive experience in the discussed scientific field and in proposed research methodology. The methodological description of the obtained results, and in particular the discussion of obtained results and possible causes, were very interesting.
Below, I’ve presented some minor editorial comments, which, in the opinion of the reviewer, will facilitate the reading of the text of the manuscript for the reader.
Abstract
1. Line: 25-26 - I would transform the sentence "Our results indicate that KRG could be a curative agent for the alleviation of asthma" into a slightly less blunt manner, less describing as "can", but instead more indicative of potential because there is still a long way to go to define the therapeutic effectiveness of KRG fully.
1. Introduction
1. Line: 47 - Shouldn't the plural form "are" be used instead of "is" in this sentence?
2. Line: 48 - 50: This sentence mentions "various studies" and only one scientific publication (number 10) is cited. Perhaps it is worth considering adding citations to at least one other publication relating to antioxidative, anti-inflammatory / immunomodulatory activities? - other, then further mentioned citations no. 11 - 16. Suggestion for consideration.
3. Line: 65 - In the last paragraph there’s lacks information about what kind of Korean Red ginseng (KRG) formula was investigated, was it an extract (if so, what was it standardized for and how was it obtained?) Or in the form of a powder suspended in the vehiculum - this detailed (methodological) information should be included in the appropriate subsection below (i.e. 2.1., 2.3., respectively).
2. Materials and Methods
2.2. Animals and Guidelines
1. If possible, please provide the consent number of Chungnam National University Animal Care and Use Committee for the planned animal experiments to be performed.
2. There is a typo in the name of "animals" in the title of the subsection - please correct the mistake.
2.3. Experimental Procedure
1. In this section I propose to add one supplementary sentence explaining why the additional procedure of aerosol nebulizations was performed, since the development of pathogenesis was previously stimulated by the fourteen-day OVA injection procedure? Only for the sake of clarity of understanding by the reader. Was it a procedure performed by the team of authors based on their own experience in this field, or on the basis of research by other researchers (then appropriate citation should be inserted)?
2.10. Statistical Analysis
1. Considering the content of the subsection - the overwhelmed title of this section is probably wrong. The description below doesn’t concern the methodology of statistical evaluation. please correct the title of this section - indicating what kind of analysis and what kind of biological material it covers - as opposed to the ones discussed above (ie: 2-7 - 2-9. ones). Rather, it’s evidently a repetition of the methodology in section 2.9. I kindly ask for a correction.
3.2. Effects of Korean Red Ginseng on Airway Hyperresponsiveness in Asthmatic Mice
1. If possible, propose to improve the sharpness Figure 1. (A) = HPLC analysis of KRG.
2. Perhaps this is an oversight by the reviewer, but I did not see in the methodological description the description of how the authors obtained the fractionation into particular types of cells in BALF = section: 2.6. Analysis of Bronchoalveolar Lavage Fluid - I think that there is a lack of a more precise description of what kind of fractions/types of cells the researchers want to obtain? Please consider whether it is worth adding at least one sentence for the sake of clarification.
3. I suggest you consider changing the description of figures - not overusing the word "treated". Perhaps you should use the term “administered” instead.
3.4. Effects of Korean Red Ginseng on Both Th2 Cytokine Production in Bronchoalveolar Lavage Fluid and IgE levels in Serum from Asthmatic Mice
1. In the description of the results, specific numerical values of the obtained IL concentrations are given, which cannot be accurately read from Figure 2. Maybe it is worthwhile to provide a summary of these figures in the form of an appropriate table (along with the specified values of statistical significance vs. what control group)? It’s a suggestion for consideration.
3.5. Effects of Korean Red Ginseng on Airway Inflammation and Mucus Production in Lung Tissue from Asthmatic mice
1. In this section, the authors provided information on the obtained differences between the groups. However, no information was given on whether these effects were statistically significant?
3.6. Effects of Korean Red Ginseng on Reactive Oxygen Species Production and Antioxidative Signaling Molecule Expression in Lung Tissue
1. Supplementing the information, legends of Figure 3A and Fig. 4 with specific descriptions, and indications in the figures relating to specific parameters discussed in this section would significantly improve the quality of reading by the reader. Please kindly consider it.
2. In the description of Fig. 4B - concerning the changes in expression of HO-1, there is too much darkened background in the blot image, against the background of the reference beta-actin. Please try to brighten this fragment of the figure because it is too much in contrast with the control beta-actin.
3.7. Effects of Korean Red Ginseng on the Expression of iNOS and NF-κB in Lung Tissue
1. In this section, similarly a smentioned above, the authors provided information on the obtained differences between the groups. However, no information was given whether these effects were statistically significant? Please add such additional information.
4. Discussion
1. Would the authors discuss in this section which of the known/determined active secondary metabolites could be, at least in part, responsible for the observed biological properties of KRG? I think that it would make the significance of the obtained research much more attractive. In this aspect, what do the obtained results look like in comparison to the results described in the scientific literature? This is just a suggestion for consideration.
Author Response
Reviewer #3:
Revision - manuscript titled: “Korean Red Ginseng Ameliorates Allergic Asthma through Sparing the Depletion of Nrf2/HO-1 Signaling”
The authors have extensive experience in the discussed scientific field and in proposed research methodology. The methodological description of the obtained results, and in particular the discussion of obtained results and possible causes, were very interesting.
Below, I’ve presented some minor editorial comments, which, in the opinion of the reviewer, will facilitate the reading of the text of the manuscript for the reader.
Abstract
- Line: 25-26 - I would transform the sentence "Our results indicate that KRG could be a curative agent for the alleviation of asthma" into a slightly less blunt manner, less describing as "can", but instead more indicative of potential because there is still a long way to go to define the therapeutic effectiveness of KRG fully.
- Thanks for your kind comments. We transform the original sentence you pointed out into a slightly less blunt manner as follows.
“Our results suggest that KRG may have the potential to alleviate asthma”
- Introduction
- Line: 47 - Shouldn't the plural form "are" be used instead of "is" in this sentence?
- Thanks for your kind comments. As you pointed out, we change “is” into “are”
- Line: 48 - 50: This sentence mentions "various studies" and only one scientific publication (number 10) is cited. Perhaps it is worth considering adding citations to at least one other publication relating to antioxidative, anti-inflammatory / immunomodulatory activities? - other, then further mentioned citations no. 11 - 16. Suggestion for consideration.
- Thanks for your kind comments. As you pointed out, we added references to research on the effects of red ginseng we mentioned in manuscript.
- Line: 65 - In the last paragraph there’s lacks information about what kind of Korean Red ginseng (KRG) formula was investigated, was it an extract (if so, what was it standardized for and how was it obtained?) Or in the form of a powder suspended in the vehiculum - this detailed (methodological) information should be included in the appropriate subsection below (i.e. 2.1., 2.3., respectively).
- Thanks for your kind comments. We totally agree with your indication. The KRG was concentrated Korean Red ginseng extract, which was produced by Korea Ginseng Corporation. We obtained KRG from the company and performed analytical and animal experiments. We amended phrase to clarify the compound hired.
- Materials and Methods
2.2. Animals and Guidelines
- If possible, please provide the consent number of Chungnam National University Animal Care and Use Committee for the planned animal experiments to be performed.
- Thanks for your kind comments. We inserted the IACUC approval number (202009A-CNU-142) in the section of 2.2. Animals and Guidelines.
- There is a typo in the name of "animals" in the title of the subsection - please correct the mistake.
- Thanks for your kind comments. We corrected the typo you pointed out.
2.3. Experimental Procedure
- In this section I propose to add one supplementary sentence explaining why the additional procedure of aerosol nebulizations was performed, since the development of pathogenesis was previously stimulated by the fourteen-day OVA injection procedure? Only for the sake of clarity of understanding by the reader. Was it a procedure performed by the team of authors based on their own experience in this field, or on the basis of research by other researchers (then appropriate citation should be inserted)?
The asthma induction model we used in present experiment is not a model used only in our laboratory, but a model commonly used. General principle of asthma mice model includes first sensitizing and then repetitively challenging mice with the same antigen. OVA sensitization on day 0 and day 14 are not continuous administration for 14 days, but sensitization to allergen twice on days 0 and 14, followed by nebulization of OVA for 21-23 days. A brief explanation about principle were added to the materials and methods section, and a reference (animal asthma model protocol) and our previous studies using it are presented below.
- References -
Haspeslagh, E.; Debeuf, N.; Hammad, H.; Lambrecht, B.N. Murine model of asthma. Methods Mol Biol 2017, 1559, 121–136.
Kim, C.Y., Kim, J.W.; Kim, J.H.; Jeong, J.S.; Lim, J.O.; Ko, J.W.; Kim T,W. Inner Shell of the Chestnut (Castanea crenatta) Suppresses Inflammatory Responses in Ovalbumin-Induced Allergic. Nutrients 2022, 12, 2067.
Ko, J.W.; Shin, N.R.; Lim, J.O.; Jung, T.Y.; Moon, C.; Kim, T.W.; Choi, J., Shin, I.S., Heo, J.D., Kim, J.C. Silica dioxide nanoparticles aggravate airway inflammation in an asthmatic mouse model via NLRP3 inflammasome activation. Regul Toxicol Pharmacol 2020, 112, 104618.
Lim, J.O.; Lee, S.J.; Kim, W.I.; Pak, S.W.; Moon, C.; Shin, I.S.; Heo, J.D.; Ko, J.W.; Kim, J.C. Titanium Dioxide Nanoparticles Exacerbate Allergic Airway Inflammation via TXNIP Upregulation in a Mouse Model of Asthma. Int J Mol Sci 2021, 22(18), 9924.
2.10. Statistical Analysis
- Considering the content of the subsection - the overwhelmed title of this section is probably wrong. The description below doesn’t concern the methodology of statistical evaluation. please correct the title of this section - indicating what kind of analysis and what kind of biological material it covers - as opposed to the ones discussed above (ie: 2-7 - 2-9. ones). Rather, it’s evidently a repetition of the methodology in section 2.9. I kindly ask for a correction.
- Thanks for your kind comments. We corrected the fatal mistakes that occurred in the process of changing the manuscript format according to Journal policy.
3.2. Effects of Korean Red Ginseng on Airway Hyperresponsiveness in Asthmatic Mice
- If possible, propose to improve the sharpness Figure 1. (A) = HPLC analysis of KRG.
- Thanks for your kind comments. We separated the Figure 1 into 2 fiigures for improving the sharpness of HPLC analysis data.
- Perhaps this is an oversight by the reviewer, but I did not see in the methodological description the description of how the authors obtained the fractionation into particular types of cells in BALF = section: 2.6. Analysis of Bronchoalveolar Lavage Fluid - I think that there is a lack of a more precise description of what kind of fractions/types of cells the researchers want to obtain? Please consider whether it is worth adding at least one sentence for the sake of clarification.
- Thanks for your kind comments. We modified the materials and methods section (2.6) for clarify the processing of the BALF used in ROS activity and cell counting. We hope that our modification will be an appropriate response to your comments.
- I suggest you consider changing the description of figures - not overusing the word "treated". Perhaps you should use the term “administered” instead.
- Thanks for your kind comments. In order to avoid repetition of the same word, “treated” was changed to “administered” when administered orally.
3.4. Effects of Korean Red Ginseng on Both Th2 Cytokine Production in Bronchoalveolar Lavage Fluid and IgE levels in Serum from Asthmatic Mice
- In the description of the results, specific numerical values of the obtained IL concentrations are given, which cannot be accurately read from Figure 2. Maybe it is worthwhile to provide a summary of these figures in the form of an appropriate table (along with the specified values of statistical significance vs. what control group)? It’s a suggestion for consideration.
- Thanks for your kind comments. As you pointed out, Figure 2 (presented as Figure 3 in revised manuscript) was resized and the numerical values were presented in the Table (Table 1).
3.5. Effects of Korean Red Ginseng on Airway Inflammation and Mucus Production in Lung Tissue from Asthmatic mice
- In this section, the authors provided information on the obtained differences between the groups. However, no information was given on whether these effects were statistically significant?
- Thanks for your kind comments. We provided information about statistical significance in manuscript. Also, size of the figure 3 (presented Figure 4 in revised manuscript) has been adjusted to make it easier to interpret the related information.
3.6. Effects of Korean Red Ginseng on Reactive Oxygen Species Production and Antioxidative Signaling Molecule Expression in Lung Tissue
- Supplementing the information, legends of Figure 3A and Fig. 4 with specific descriptions, and indications in the figures relating to specific parameters discussed in this section would significantly improve the quality of reading by the reader. Please kindly consider it.
- Thanks for your kind comments. We resized figures 3A and 4 (presented Figure 4A and Figure 5 in revised manuscript) for better interpretation, and brief descriptions of the positive sign of each figure has been inserted in the manuscript (3.5 and 3.6)
- In the description of Fig. 4B - concerning the changes in expression of HO-1, there is too much darkened background in the blot image, against the background of the reference beta-actin. Please try to brighten this fragment of the figure because it is too much in contrast with the control beta-actin.
- Thanks for your kind comments. As you pointed out, we adjusted the brightness of the band image of the HO-1 (presented Figure 5B in revised manuscript).
3.7. Effects of Korean Red Ginseng on the Expression of iNOS and NF-κB in Lung Tissue
- In this section, similarly a smentioned above, the authors provided information on the obtained differences between the groups. However, no information was given whether these effects were statistically significant? Please add such additional information.
- Thanks for your kind comments. We provided information about statistical significance in manuscript.
- Discussion
- Would the authors discuss in this section which of the known/determined active secondary metabolites could be, at least in part, responsible for the observed biological properties of KRG? I think that it would make the significance of the obtained research much more attractive. In this aspect, what do the obtained results look like in comparison to the results described in the scientific literature? This is just a suggestion for consideration.
- We totally agree with reviewers’ indication. We amended discussion part by adding comments, which contained potential metabolite candidate and difficulties in indicating leading active compound from KRG.

Round 2
Reviewer 1 Report
The authors answered the questions posed by the reviewers. The quality of the figures improved considerably and current manuscripts from the same scientific area were included in the discussion.
Author Response
Reviewer #1: The authors answered the questions posed by the reviewers. The quality of the figures improved considerably and current manuscripts from the same scientific area were included in the discussion.
- Thank for your kind consideration in the revision process.

Reviewer 2 Report
Although the authors have modified the manuscript according to the suggestions, the Western blot results are not satisfying.
On Western blots, molecular weights should be indicated. Also, blots with protein markers should be in supplemental data.
- Thanks for your kind comments. We inserted the molecular weights of proteins investigated in the present study. In addition, the original Western blot band submitted in the first submission process were inserted as supplemental data.
1st there is no visible protein weight marker photo, for this one photo of the membrane without antibody, but with a visible marker (it could be stained protein markers or ponceau S that makes the marker visible)
How do authors explain the unspecific bands for all blots? NRF2 and HO-1 are only one band without so many unspecific bands that the authors presented.
- Thanks for your kind comments. It is thought that there are various reasons for the occurrence of unspecific bands for western blot. According to our experimental experience so far, the 5 major reasons for unspecific band are as follows: inadequacy of tissue sampling process, buffer composition, pH, antibody purity, and blocking. These reasons may cause not only the generation of an unspecific band but also the splitting of the one target band into several strands at the expected molecular weight. Also, while there are antibodies that show these phenomena well, there are antibodies that show only the target band well (actin or HO-1 you pointed out). In the present study, unspecific bands appeared because all these factors could not be controlled.
However, it is not considered inappropriate to interpret a band that appears clearly at the molecular weight point suggested by the manufacturer as a target protein. We hope that our answer will be an appropriate response to your comments.
2nd There is a problem with finding a good antibody which clearly hit the target. If the purchased antibody shows bands as seen by the authors, this s a clear indication that the antibody is not good and specific. Buffer composition, pH, and blocking are actually under the control of the person who performs the experiments, and these conditions should be optimized for each antibody. All purchased antibodies should be pure, if you doubt antibody purity you should make a complaint to the manufacturer. Therefore, these blots are too unspecific to trust in the results.
In addition, Nrf2 is a specific protein, further reading; Lau, A., Tian, W., Whitman, S. A., & Zhang, D. D. (2013). The predicted molecular weight of Nrf2: it is what it is not. Antioxidants & redox signaling, 18(1), 91–93. https://doi.org/10.1089/ars.2012.4754
Author Response
Reviewer #2: Although the authors have modified the manuscript according to the suggestions, the Western blot results are not satisfying.
1st there is no visible protein weight marker photo, for this one photo of the membrane without antibody, but with a visible marker (it could be stained protein markers or ponceau S that makes the marker visible)
- Thanks for your kind comments. Molecular weight of each protein was added to the main figure data, and most protein markers of original image band not shown in the main manuscript can be found in supplement figure 1.
2nd There is a problem with finding a good antibody which clearly hit the target. If the purchased antibody shows bands as seen by the authors, this s a clear indication that the antibody is not good and specific. Buffer composition, pH, and blocking are actually under the control of the person who performs the experiments, and these conditions should be optimized for each antibody. All purchased antibodies should be pure, if you doubt antibody purity you should make a complaint to the manufacturer. Therefore, these blots are too unspecific to trust in the results.
In addition, Nrf2 is a specific protein, further reading; Lau, A., Tian, W., Whitman, S. A., & Zhang, D. D. (2013). The predicted molecular weight of Nrf2: it is what it is not. Antioxidants & redox signaling, 18(1), 91–93. https://doi.org/10.1089/ars.2012.4754
- Thanks for your kind comments. We totally agree with your comments. However, as mentioned in the first revision, the currently presented data is the best for interpreting the original blot image presented in the supplemental figure 1. We would appreciate it if you could understand our current situation. We will reconsider the purity of our antibody and change the experimental conditions to achieve the best results. Also, we will refer to the paper you suggested and use it as a good data for further research on Nrf2.
